# Genomic Characterisation of UFJF_PfDIW6: A Novel Lytic *Pseudomonas fluorescens*-Phage with Potential for Biocontrol in the Dairy Industry

**DOI:** 10.3390/v14030629

**Published:** 2022-03-17

**Authors:** Humberto Moreira Hungaro, Pedro Marcus Pereira Vidigal, Edilane Cristina do Nascimento, Felipe Gomes da Costa Oliveira, Marco Túlio Pardini Gontijo, Maryoris Elisa Soto Lopez

**Affiliations:** 1Departamento de Ciências Farmacêuticas, Faculdade de Farmácia, Universidade Federal de Juiz de Fora (UFJF), Juiz de Fora 36036-900, MG, Brazil; edilanemv@outlook.com (E.C.d.N.); lipygomes12@yahoo.com.br (F.G.d.C.O.); 2Núcleo de Análise de Biomoléculas (NuBioMol), Campus da UFV, Universidade Federal de Viçosa (UFV), Viçosa 36570-900, MG, Brazil; pedro.vidigal@ufv.br; 3Departamento de Genética, Evolução, Microbiologia e Imunologia, Instituto de Biologia, Universidade Estadual de Campinas (UNICAMP), Campinas 13083-872, SP, Brazil; m264546@dac.unicamp.br; 4Departamento de Engenharia de Alimentos, Universidade de Córdoba (UNICORDOBA), Córdoba 230002, Colombia

**Keywords:** bacteriophage genome, *Pseudomonas*, comparative analysis, *Podoviridae*, taxonomy

## Abstract

In this study, we have presented the genomic characterisation of UFJF_PfDIW6, a novel lytic *Pseudomonas fluorescens*-phage with potential for biocontrol in the dairy industry. This phage showed a short linear double-stranded DNA genome (~42 kb) with a GC content of 58.3% and more than 50% of the genes encoding proteins with unknown functions. Nevertheless, UFJF_PfDIW6’s genome was organised into five functional modules: DNA packaging, structural proteins, DNA metabolism, lysogenic, and host lysis. Comparative genome analysis revealed that the UFJF_PfDIW6’s genome is distinct from other viral genomes available at NCBI databases, displaying maximum coverages of 5% among all alignments. Curiously, this phage showed higher sequence coverages (38–49%) when aligned with uncharacterised prophages integrated into *Pseudomonas* genomes. Phages compared in this study share conserved locally collinear blocks comprising genes of the modules’ DNA packing and structural proteins but were primarily differentiated by the composition of the DNA metabolism and lysogeny modules. Strategies for taxonomy assignment showed that UFJF_PfDIW6 was clustered into an unclassified genus in the *Podoviridae* clade. Therefore, our findings indicate that this phage could represent a novel genus belonging to the *Podoviridae* family.

## 1. Introduction

*Pseudomonas* is an important group of bacteria widely distributed in several ecological niches, including plants, soil, water, animals and humans [1]. This genus includes species saprophytes, commensals, and pathogens to plants, insects, animals, and humans, with which some have the potential for biotechnological applications [2]. For example, in the dairy industry, *Pseudomonas* spp., mainly *P. fluorescens*, are common contaminants associated with the spoilage of milk and dairy products [3,4]. The growth of *Pseudomonas* in milk has been long associated with the production of peptidases and lipases, leading to rancid, bitter flavour, off-flavour, milk heat-stability loss, gelation of UHT milk, and reduction of yield in cheese manufacturing [5,6,7].

Bacteriophages have emerged as a promising alternative to bacterial control in the food chain [8]. Although bacteriophages have long traditionally been considered enemies for the dairy industry, especially in fermented dairy products, they have been recognized as a potential solution for controlling pathogenic and spoilage bacteria in this environment [9,10,11]. Examples of biocontrol in the dairy industry have included the use of phages against *Staphylococcus aureus* in different types of milk and cheese [12,13,14,15]; *Listeria monocytogenes* in chocolate milk and cheese [16,17,18,19,20], pathogenic *Escherichia coli* in raw milk and UHT milk [21,22] *Salmonella* in milk and cheese [23,24], and *Pseudomonas* in raw milk and UHT milk [25,26,27].

Ideally, bacteriophages exhibit species specificity when used to prevent the growth of undesired bacteria in milk that act as spoilage agents. Bacteriophages that infect lactic acid bacteria represent a great concern in the dairy industry, which might result in production delays, waste of ingredients and raw materials, lower quality of the fermented product, growth of spoilage and/or pathogenic microbes, or even total loss in the processing plant [28]. Therefore, the host range of bacteriophages must be assessed to ensure application safety.

*P. fluorescens*-infecting phages had been isolated, characterised, and evaluated to the ability to control bacterial contamination in different environments, including milk and dairy products and biofilms on abiotic surfaces [29,30,31,32,33,34,35]. In our previous study, phage UFJF_PfDIW6 isolated from dairy industry wastewater could control *P. fluorescens* growth and reduce casein hydrolysis in raw milk. UFJF_PfDIW6 was also shown as highly specific to *P. fluorescens* UFV 041. Using this host, we obtained a titre of 9.7 ± 0.18 log PFU/mL, a latent period of 115 min, a burst size of 145 PFU/infected cell, a thermal resistance at 63 °C or 72 °C for 30 min, and stability at pH 5 to 11 [27].

UFJF_PfDIW6 was assigned as a putative member of the *Podoviridae* family due to its morphological features. In comparison to phages infecting the other species of the genus *Pseudomonas* (host taxid: 286; *n* = 731), only a small number of phage species infecting the *P. fluorescens* group (host taxid: 136843; *n* = 37) have been sequenced and are available at the NCBI Virus database (https://www.ncbi.nlm.nih.gov/labs/virus; accessed on 26 October 2021). These *P. fluorescens* phages were assigned to the *Autographiviridae* family (*n* = 17), *Myoviridae* (*n* = 5), *Podoviridae* (*n* = 5), *Siphoviridae* (*n* = 4), *Schitoviridae* (*n* = 1), and one is still unclassified. All these families are classified into the *Caudovirales* order, which was proposed in 1998 by Hans Wolfgang Ackermann to comprise all tailed phages [36]. The advent of the genomic era revealed a much higher genomic diversity for this order, and the paraphyly observed in phylogenetic studies has prompted the International Committee on the Taxonomy of Viruses (ICTV) to disentangle *Caudovirales*’ taxonomy by defining new genome-based families [36].

Given these findings, the sequencing of the UFJF_PfDIW6 genome and its comparison to other *Pseudomonas* phages are essential to solve its taxonomy and provide information for its future commercial use as biocontrol agents in the dairy industry.

Here, we report the genome sequence of *Pseudomonas fluorescens*-phage UFJF_PfDIW6 and a taxonomy proposal to comprise its genomic features, based on the functional prediction of proteins, host spectrum, and comparative analysis to other *Pseudomonas* phages.

## 2. Materials and Methods

### 2.1. Phage Isolation and Propagation

The UFJF_PfDIW6 phage, previously isolated from dairy industry wastewater, was propagated as described by Sambrook and Russell [37] at 30 °C in TSB medium using *P. fluorescens* UFV 041 (host bacterium), kindly provided by the Culture Collection of the Laboratory of Food Microbiology, Department of Microbiology, Federal University of Viçosa, Brazil.

### 2.2. Phage DNA Extraction and Sequencing

The UFJF_PfDIW6 phage was propagated in TSB medium containing *P. fluorescens* UFV 041 in exponential phase at 30 °C for 24 h, centrifuged (10 min, 2202× *g*, 4 °C), and filtered (0.22 μm pore size). Phage suspension was incubated with nuclease mix (0.25 mg/mL RNAse A, 0.25 mg/mL DNAse I, 150 mM NaCl, 50% *v*/*v* glycerol) at 37 °C for 30 min followed by 25 °C for 1 h to remove bacterial DNA and RNA. Then, the phage particles were precipitated using precipitant suspension (20% *w*/*v* PEG 8000, 1.76 M NaCl), centrifuged (20 min, 12,000× *g*, 4 °C) and resuspended in SM buffer (50 mM Tris–HCl (pH 7.5), 0.1 M NaCl, 8 mM MgSO_4_ × 7H_2_O). After that, phage DNA was extracted using the Wizard DNA Clean-Up kit with modifications as proposed by Summer [38]. The degradation, purity, and concentration of DNA extracted were evaluated by 1% agarose gel electrophoresis, spectrophotometry (OD260/OD280) using a NanodropTM (Thermo Fisher Scientific, Waltham, MA, USA), and fluorimetry using a Qubit 2.0 fluorometer (Life Technologies (Carlsbad, CA, USA), Invitrogen (Waltham, MA, USA), respectively.

UFJF_PfDIW6 genome library construction was performed using the NEBNext Ultra II DNA Library Prep Kit for Illumina (New England Biolabs, Ipswich, MA, USA) according to the manufacturer’s instructions and sequenced using the Illumina NovaSeq 6000 platform with a read length of 2 × 150 bp by GenOne Biotech (Rio de Janeiro, RJ, Brazil).

### 2.3. Quality Control, Trimming and Filtering

The quality of sequencing data was assessed using FASTQC version 0.1.16 (https://github.com/s-andrews/FastQC). Adapter sequences were detected and removed from sequencing data using the “auto-detection” setting of TrimGalore version 0.6.7 [39]. Then, paired reads were trimmed for quality and filtered for length using Trimmomatic version 0.39 [40] by selecting the following parameters: HEADCROP:10, CROP:135, SLIDINGWINDOW:4:20, and MINLEN:50.

### 2.4. Genome Assembly, Gene Prediction and Protein Annotation

The ab initio assembly of the phage genome was performed using the “careful” option of SPAdes version 3.14.1 [41] and all odd k-mers between 21 and 127. Then, paired reads were mapped to the assembly using the BWA-MEM algorithm of BWA version 0.7.17 [42] and insert size metrics were collected using Picard toolkit version 2.26.2 (https://github.com/broadinstitute/picard) to perform an additional step of scaffolding using SSPACE version 3.0 [43]. The final assembly was evaluated using assembly-stats version 1.0.1 (https://github.com/sanger-pathogens/assembly-stats), and the coverage of scaffolds was calculated using BBMap version 38.76 (https://sourceforge.net/projects/bbmap).

The UFJF_PfDIW6 genome was scanned for terminal repeats using PhageTerm version 4.0.0 [44] with default parameters. Genes were predicted using Prokka version 1.14.6 [45] by selecting the following parameters: compliant, kingdom: viruses, gcode: 11, cdsrnaolap, and E-value: 1e^−10^. Putative promoters were identified using the PhagePromoter (https://galaxy.bio.di.uminho.pt/) [46], by selecting the following parameters: threshold: 0.5, phage family: *Podoviridae*, host bacteria genus: *Pseudomonas*, and phage type: virulent. Rho-independent terminators were identified using ARNold [47] (http://rssf.i2bc.paris-saclay.fr/toolbox/arnold/) with default parameters. The host spectrum of UFJF_PfDIW6 was predicted using HostPhinder server version 1.1 (http://cge.cbs.dtu.dk/services/HostPhinder) [48], and its lifestyle was predicted using BACPHLIP version 0.9.3-alpha [49].

UFJF_PfDIW6 genome and predicted proteins were submitted to sequence similarity searches using BLAST [50], considering an E-value threshold ≤1e^−10^ for selecting significant alignments. The alignments were further evaluated by query coverage, identity and or similarity.

The genome sequence was aligned to viruses genomes (taxonomy ID 10239) available at the NCBI Nucleotide collection (nt) database using the BLASTn tool of the web BLAST server (https://blast.ncbi.nlm.nih.gov/). A search for related prophages was also performed by aligning the phage genome to bacteria genomes (taxonomy ID 2). PHASTER (PHAge Search Tool Enhanced Release) webserver (https://phaster.ca/) [51,52] was used to screen the prophages in the bacteria genomes that showed significant alignments with the UFJF_PfDIW6 genome.

The protein sequences were aligned to viral proteins available at the NCBI Virus database (26 October 2021; Taxonomy ID 10239) using the BLASTp tool of BLAST version 2.12.0. Only significant alignments with overall similarity (similarity × coverage) ≥ 40% were considered for annotation of UFJF_PfDIW6 proteins.

In addition, protein sequences were aligned to profiles with hidden Markov models (HMMs) available at the Virus Orthologous Groups (VOG) database (https://vogdb.csb.univie.ac.at/, release vog207) using HMMER version 3.3.2 [53] by selecting the following parameters for significant alignments: E-value ≤ 1e^−10^ and bias score = 0.

### 2.5. Endolysin Screening

Endolysins were screened using HmmerWeb version 2.41.2 (https://www.ebi.ac.uk/Tools/hmmer/) [54] and Protein family database (Pfam) (http://pfam.xfam.org/) [55] applying default parameters. Signal peptides and signal-arrest–release domains were characterized as proposed by [56] and Oliveira et al. [57]. Signal peptides were predicted using SignalP version 5.0 [58] and PrediSi version 1.0 [59] against a Gram-negative database. Transmembrane regions were predicted using SOSUI version 1.1 [60], TMHMM version 2.0 [61], Phobius version 1.01 [61], and Topcons version 1.0 [62].

### 2.6. Comparative Genomics

The UFJF_PfDIW6’s genome was compared to other genomes of phages and prophages that showed significant alignments in the similarity searches. The selected genomes were aligned using the progressive Mauve algorithm [63] of Mauve version 2.4.0 (http://darlinglab.org/mauve/mauve.htm) to identify conserved locally collinear blocks (LCBs). The synteny and genes’ modular organisation of phages and prophages genomes were evaluated using Clinker version 0.0.23 [64].

### 2.7. Taxonomy Assignment and Phylogenetic Analysis

UFJF_PfDIW6’s taxonomy assignment was evaluated using the VIRIDIC web server (http://rhea.icbm.uni-oldenburg.de/VIRIDIC/) [65] and were compared to its genome sequence with the genomes of 130 reference species of the *Podoviridae* family which are recognised by the International Committee on Taxonomy of Viruses (ICTV) (https://talk.ictvonline.org/taxonomy/, release 2020). UFJF_PfDIW6’s genome was also compared to the genomes that showed significant results in similarity searches. We considered a genomic similarity threshold of 95% for species assignment and 70% for the genus.

Phylogenetic analysis was performed by selecting terminase large subunit (TerL) amino acid sequences from UFJF_PfDIW6 and other TerL sequences of *Pseudomonas* phages available at the NCBI RefSeq database (https://www.ncbi.nlm.nih.gov/refseq/, accessed on 26 October 2021). TerL sequences were aligned by MAFFT version 7.487 [66] using a local pairwise alignment with 100,000 iterations of parameter refinements to improve the alignment. The alignment was trimmed using the “automated1” heuristic method of Trimal version 1.4.1 [67], and the best-fit amino-acid substitution model was selected according to Bayesian Information Criterion (BIC) using Modeltest-NG version 0.1.6 [68]. The Phylogenetic tree was inferred by Maximum Likelihood (ML) using RAxML-NG version 1.0.3 [69]. The Majority Rule (MR) consensus tree was calculated with 1000 bootstrap replicates and then visualised using Figtree version 1.4.4 (https://github.com/rambaut/figtree).

## 3. Results and Discussion

The UFJF_PfDIW6 phage was previously isolated from dairy industry wastewater and described as showing great potential to be applied for *Pseudomonas* biocontrol and prevent spoilage of raw milk stored under refrigeration [27]. The plaque morphology of UFJF_PfDIW6 on its propagating host *P. fluorescens* UFV 041, virion morphology, and a summary of lifecycle characteristics are shown in Appendix A.

### 3.1. UFJF_PfDIW6’s Genome Features

The genome of the UFJF_PfDIW6 phage was assembled as a circular sequence without terminal repeats, which represents its replicative intermediate. This genome was sequenced with 35,802 fold coverage and is available at the GenBank database (https://www.ncbi.nlm.nih.gov/genbank/) under accession OM418631.

The UFJF_PfDIW6 genome comprises 42,322 bp of linear double-stranded DNA with a GC content of 58.3%. Its sequence has a bidirectional organisation with a gene density of 1.38 gene/Kb, including 58 coding DNA sequences (CDSs), and was opened to the small terminase subunit gene (TerS; g01) (Figure 1 and Appendix A). Thirty-eight CDSs encode proteins similar to other phage proteins, while nineteen CDSs encode proteins similar to bacteria’s proteins that possibly represent prophages. Only one CDS encodes a protein that did not significantly align to any sequence. Thirty-one proteins had a predicted function, while twenty-seven were annotated as hypothetical proteins. Twenty-five Rho-independent terminator sequences (Appendix A), four promoter sequences recognised by host’s RNA polymerase (RNAP) and one by phage’s RNAP (Appendix A) were also found in the UFJF_PfDIW6 genome. In addition, no evidence of RNAP was found among predicted proteins, even among profiles of the Virus Orthologous Groups database (Appendix A), and no tRNA genes were found, which suggest that UFJF_PfDIW6 phage could be highly dependent on host translation machinery.

Despite that more than 50% of proteins had no predicted functions, UFJF_PfDIW6’s genome can be organised into five functional modules: (A) DNA packaging; (B) structural proteins; (C) DNA metabolism (replication, processing and modification); (D) lysogenic; and (E) host lysis (Figure 1).

Three major genes encode small and large subunits of terminase, and portal proteins are generally identified in the DNA packaging module of phages’ genomes. Terminase proteins are responsible for recognition and cleavage of viral DNA and its translocation into a preformed empty phage capsid by ATP hydrolysis [70,71]. Portal proteins also play a crucial role in virion assembly, DNA packaging, and DNA delivery in phages of the Caudovirales order. They serve as a portal for phage DNA passage during DNA packaging and ejection [72]. UFJF_PfDIW6’s genome is opened by two genes that encode the terminase small subunit (TerS; g01) and large terminal subunit (TerL; g02). The third gene (g03) encodes a 117-aa protein that did not significantly align to any sequence available in the selected databases, but we speculate that it encodes a portal protein.

The structural module contains nine genes that encode virion’s structural proteins such as head-to-tail-connector (g04), stabilisation protein (g06), capsid protein (g07), tail tubular proteins A (g10), tail tubular proteins B (g11 and g12), internal virion proteins (IVPs) A and B (g13 and g14) and tail spike protein (g17). This module also has two proteins, a structural protein (g08), and a structural lysozyme (g15). Three genes (g05, g09 and g16) that encode proteins with unknown functions are located among the structural genes and are possibly virion structural proteins.

Capsid proteins are essential structural components of the virion and protect the viral genome from the harsh outer environment. The tail structures form a complex macromolecular machine responsible for recognition and penetration of the host cell and channel formation for phage genome injection [73]. Two-tail tubular proteins (A and B) are frequently identified in the genome of Podoviridae phages. Tail tubular A forms the attachment for the virion’s side fibres and mediates the initiation of the infection process working similar to a conformational switch. Side fibre engagement causes the release of capsid contents. Tail tubular B associates directly below the protein A in a nozzle-like structure, extending the tube through which the DNA will pass [74].

Other structural proteins vital in the viral infection process are the Internal Virion Protein (IVPs), which form a transient tube crossing the cell wall to the cellular membrane and protect the viral genome from periplasmic nucleases [75]. IVPs are sometimes associated with lysozymes [74], which have lytic transglycosylase activity and break the peptidoglycan layer of the host cell during the phage’s infection process.

Seven genes were included in the DNA metabolism module for encoding proteins displaying functions related to replication, regulation and modification; they are topoisomerase (g18), endonuclease (g29), DnaC DNA replication protein (g45), recombination protein NimB-like (g48), recombination protein NinG (g50), antitermination Q protein (g51), and HNH homing endonuclease II (g54).

DNA topoisomerases are enzymes that play a crucial role in DNA metabolism, as they can solve DNA topological problems and enable processes such as DNA replication, transcription, and recombination [76]. DnaC proteins are known to recruit the helicase protein to the initiator complexes into replication origins in both phages and bacteria [77]. HNH endonucleases are large group proteins that have been identified from various organisms, including bacteriophages and bacteria. They have been related to DNA packaging and described as a cofactor associated with terminases for many phages [78,79].

Our previous study assessed eight strains of five *Pseudomonas* species, and the UFJF_PfDIW6 phage was host specific [27]. UFJF_PfDIW6 had a typical lytic phage behaviour during all experimental assays by showing clear lysis plaques on the lawn of *P. fluorescens* UFV 041 host since its purification (Appendix A), high titres after successive rounds of propagation, and infectivity after storage under both refrigeration and freezing [27]. However, UFJF_PfDIW6 was predicted as having a temperate lifestyle (*p* = 0.90) by the BACPHLIP algorithm. A detailed analysis of its genome revealed a typical lysogeny module containing genes that encode two Arc family DNA-binding proteins (g19 and g20), antirepressor (g21), integrase (g22), TolA protein (g31), Cro/CI family transcriptional regulator (g41) and CII regulatory (g43) proteins (Figure 1). There is no evidence of gene encoding excisionase in UFJF_PfDIW6’s genome, an essential enzyme in the excision of prophage from the bacterial chromosome to establish the phage’s lytic cycle. However, some authors reported that, eventually, integrase could catalyse both integration and excision reactions, regardless of the presence of an excisionase [80,81].

During lysogeny, the phage genome is integrated into the bacterial chromosome as a prophage, which will be replicated through new host generations [82,83]. The switch between the lytic and lysogenic cycle depends on the presence and regulation of some genes of the phages’ genome, such as genes that encode CII regulatory protein and Cro/CI-like repressor. CII regulatory protein activates the transcription of both CI and integrase leading to lysogeny [84]. Conversely, a Cro-like repressor is a repressor of CI, which leads the phage to the lytic cycle [85].

Many genes encoding hypothetical proteins were annotated among genes of the lysogeny module, and some critical genes for the complete functioning of this module were not found, which suggests that the UFJF_PfDIW6 genome has an incomplete module, or an environmental factor could explain its lytic behaviour. In addition to crucial genes of the lysogeny module, both host incubation temperature and multiplicity of infection (MOI) can affect the lysis–lysogeny decision in the phage infection. For example, in the Enterobacteria Phage Lambda, the lysis–lysogeny decision occurs by cII protein accumulation, which activates transcription of the cI gene and the integrase gene, culminating in lysogen formation [86]. The accumulation of the CII protein depends on no activation of a protease encoded by the host hfl gene, which is inhibited by poor host growth conditions. Similarly, high MOI increases CIII protein, a proteolysis inhibitor of cII protein. Changes in the size and appearance of lysis plaques were observed when host infections by UFJF_PfDIW6 occurred at 30 or 37 °C, following the same conditions previously described [27].

Two genes that encode phage holin (g55) and endolysin (g56) were classified as being components of the host lysis module (Figure 1). UFJF_PfDIW6 was highly specific regarding the host range in previous tests, infecting only the host bacterium among 23 bacterial strains belonging to nine genera [27]. However, this phage reduced the bacterial count of naturally contaminated raw milk, indicating the infection ability of other bacterial species.

The host range of UFJF_PfDIW6 was predicted using its genome to complement previously reported experimental results. Computational tools based on whole phage genome sequences analysis have been frequently used to predict host range [87]. HostPhinder compares k-mers between the query phage genome and the genomes of reference phages with known hosts [48]. No bacterial species from the HostPhinder’s database were predicted as a target to be infected by UFJF_PfDIW6 (Appendix A), contrasting to the experimental evidence that confirmed its effectiveness in reducing raw milk microbiota. This divergence could be explained by limitations of HostPhinder’s database regarding its completeness, with a limited number of phage reference genomes and bacterial hosts [48].

Beyond host spectrum analysis, the analysis of the host lysis module revealed some aspects about the interaction of UFJF_PfDIW6 with its bacterial hosts. Genes encoding holin (g55) and endolysin (g56) were annotated in this module. The holin belongs to the bacteriophage holin family, superfamily II-like (Pfam accession PF16082; E-value: 2.1e^−15^). The PF16082 Pfam is commonly found in the genomes of *Pseudomonas* bacteriophages, as well as other holin-family proteins: PF05106, PF05449, PF10746, PF13272, PF16080, PF16082, PF16083, PF16085, and PF16931 [88].

The endolysin belongs to the phage lysozyme family (Glycoside hydrolase family 24; PF00959; E-value: 3.7e^−4^). Among the muramidase superfamily of endolysins, PF00959 is the most common Pfam domain in *Pseudomonas* phages endolysins [88]. In addition, the endolysin contains a signal peptide with a cleavage site at residue 26 (likelihood: 0.9005), and no transmembrane helices were found on its sequence. These results suggest that predicted endolysin contains a canonical signal peptide compatible with the Sec-mediated transport and no SAR (signal-arrest-release) domain.

The canonical lysis module of bacteriophages includes endolysins and holins [89]. Endolysins are responsible for peptidoglycan (PG) disruption after holins form pores in the cytoplasmatic membrane. These pores provide PG access to endolysins [90]. In the case of SP-endolysins, holins also control the depolarization of the cell membrane and trigger the activation of secreted endolysins [90,91]. This is the case for the UFJF_PfDIW6 phage.

Some bacteriophages also encode pinholins and, in the case of bacteriophages that infect Gram-negative bacteria, spanins [89]. Spanins form junctions between the inner and the outer membrane of Gram-negative bacteria to promote the release of the new viral particles after viral infection [92]. No spanin homolog was found in the genome of the UFJF_PfDIW6 phage.

### 3.2. Comparative Genomics

Similarity searches revealed that the UFJF_PfDIW6’s genome is significantly different from other viral genomes available at the NCBI GenBank database, displaying a maximum coverage of 5% among all alignments (Appendix A). This intriguing result led to an investigation into whether its genome could be similar to putative prophages integrated into bacterial genomes once bacteria can harbour several phages that are still uncharacterised [93]. UFJF_PfDIW6’s genome showed higher sequence coverages when aligned with genomes from *Pseudomonas* species, ranging from 38% to 49% (Appendix A). In addition, all bacteria’s genomic regions that were targets of the alignments contain putative prophages classified as intact by the PHASTER.

Five prophage genomes and two genomes of *Pseudomonas* phages (AF and HU1) were selected and compared to UFJF_PfDIW6’s genome (Figure 1 and Appendix A). The selected genomes were re-opened to the gene that encodes terminase small subunit (TerS). Comparative genomic analysis revealed that they share conserved locally collinear blocks (LCBs) comprising genes of DNA packing and structural proteins modules. UFJF_PfDIW6’s genome shares *Pseudomonas* prophage’s one LCB, which includes two genes of the host lysis module. The genes included in these LCBs are syntenic and encode proteins related to virion assembly and its interaction with host bacteria. Based on these genomic features, some insights about the host spectrum of UFJF_PfDIW6 phage can be considered, such as its ability to infect other *Pseudomonas* species.

Among structural proteins, tail spike protein was suggested as being the only virion component of *Pseudomonas* phage AF that determines its host specificity [94]. The protein encoded by gene g17 of UFJF_PfDIW6’s genome only showed significant similarity with a tail spike protein from a metagenome-assembled genome (MAG) sequence classified as *Siphoviridae* sp. (Appendix A). However, UFJF_PfDIW6’s tail spike protein also showed a high similarity with hypothetical proteins of putative prophages of *Pseudomonas* sp. HN8-3 (NCBI accession WP_228308670.1; 95% of similarity), *P. fragi* (WP_169907497.1; 81%), *P. lundensis* (WP_070413385.1; 81%) and *P. lactis* (WP_057976804.1; 82). These findings suggest a broader host range spectrum for UFJF_PfDIW6 phage and corroborate the experimental evidence about its ability in reducing raw milk microbiota [27].

In contrast to the structural and host lysis modules, the genomic region, including the modules of DNA metabolism and lysogeny, is the primary point of differentiation among all analysed genomes. This region corresponds to 44% of UFJF_PfDIW6’s genome and is delimited by genes g18 (position 22,019) and g54 (position 40,821), comprising most of the genes that encode proteins with unknown function (Appendix A).

### 3.3. Taxonomic Assignment

Because comparative genomics revealed conserved regions among UFJF_PfDIW6’s genome and other phages’ genomes, the question is its taxonomic classification.

The genomic similarity matrix calculated by VIRIDIC showed that UFJF_PfDIW6’s genome is different from all reference genomes of species classified in the *Podoviridae* family listed in the current release of the International Committee on Taxonomy of Viruses (ICTV). UFJF_PfDIW6’s genome was placed in different species and genus clusters from these genomes (Appendix A). Among RefSeq genomes, the genome of *Salmonella virus Epsilon* 15, from the *Uetakevirus* genus, showed the highest similarity with 5.54%, far from the genomic similarity threshold of 95% for species assignment and 70% for the genus. The genome of prophage identified in *Pseudomonas* sp. HN8-3 with 47.81% was the most similar among the genomes presented in Figure 1. In addition, the genomes of *Pseudomonas* phage AF and *Pseudomonas* phage HU1 showed 6.83% and 10.22% of genomic similarity, respectively (Appendix A).

A phylogenetic hypothesis was inferred using the terminase large subunits (TerL) protein sequences from UFJF_PfDIW6 and other *Pseudomonas* phages as evolutionary markers to solve its taxonomic assignment (Figure 2). All genera were included in monophyletic clades in the phylogenetic tree, supported by high bootstrap values and reinforcing the TerL as an excellent phylogenetic marker for phages’ genus delimitation.

The UFJF_PfDIW6 phage was clustered with *Pseudomonas* phage AF and *Pseudomonas* phage HU1, which displayed conserved regions in comparative genomic analysis representing a monophyletic cluster of the unclassified genus in the *Podoviridae* clade. A genomic similarity matrix was also calculated for these *Pseudomonas* phages and revealed that even phages previously classified in the same genus could be placed in different genus clusters by VIRIDIC (Figure 2). In addition, UFJF_PfDIW6 phage, *Pseudomonas* phage AF and *Pseudomonas* phage HU1 were also placed in different genus and species clusters.

The viral proteomic tree (ViPTree), including the reference genomes of the *Podoviridae* family, also clustered the genomes of UFJF_PfDIW6 phage, *Pseudomonas* phage AF and *Pseudomonas* phage HU1 in a distinct monophyletic cluster of unclassified viruses (Figure 3), such as that observed in the phylogenetic tree based on TerL proteins of *Pseudomonas* phages. Taken together, all evidence presented here reinforces that the UFJF_PfDIW6 phage is representative of a novel genus of the *Podoviridae* family, such that *Pseudomonas* phage AF and *Pseudomonas* phage HU1 are also single species of novel genera. These findings are in line with the need for taxonomic revision of the order *Caudovirales* through genome-based approaches [36].

## 4. Conclusions

In this study, we report a taxonomic proposal for *Pseudomonas* phage UFJF_PfDIW6 to represent a novel genus of the *Podoviridae* family. The UFJF_PfDIW6’s genome comprises 42,322 bp of linear double-stranded DNA containing 58 genes organised into five functional modules. Despite UFJF_PfDIW6 showing a lytic behaviour in the in vitro assays on the host lawn, a set of genes related to lysogeny was annotated on its genome. Comparative genomic analysis showed that UFJF_PfDIW6 shares, with other *Pseudomonas* phages, conserved locally collinear blocks (LCBs) comprising genes of DNA packing and structural protein modules related to virion assembly and host recognition. However, UFJF_PfDIW6’s sequence is distinct from other genomes of virulent phages and prophages, having genomic similarities below the recommended thresholds for genus and species assignments. Phylogenetic analysis, using the terminase large subunit (TerL) protein sequences and viral proteomic tree, clustered UFJF_PfDIW6 into a monophyletic cluster of unclassified viruses, being a single representative of a novel genus in the *Podoviridae* clade.

## Figures and Tables

**Figure 1 viruses-14-00629-f001:**
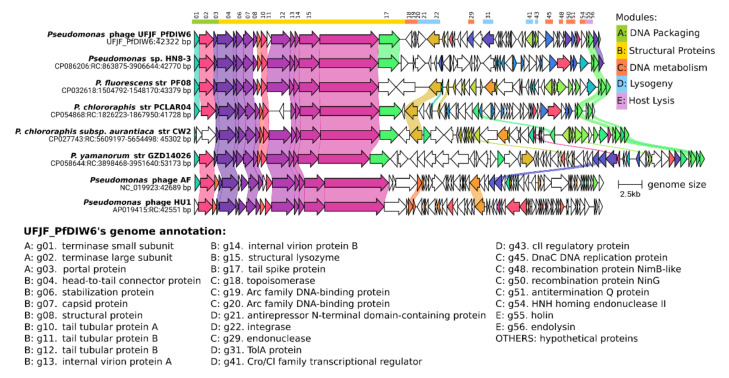
Genomic features of *Pseudomonas* phage UFJF_PfDIW6. UFJF_PfDIW6’s genome is shown above all others. Its sequence has a bidirectional organisation with 58 genes (represented by arrows) organised into five functional modules, represented by the coloured bars and indicated in the genes’ annotation list. The numbers above the arrows indicate the genes with predicted functions, shown in the annotation list. The arrow’s colours represent the gene clusters identified by Clinker, which encode similar proteins. The lines connecting the arrows represent gene-encoding proteins that share sequence identity more significant than 40%.

**Figure 2 viruses-14-00629-f002:**
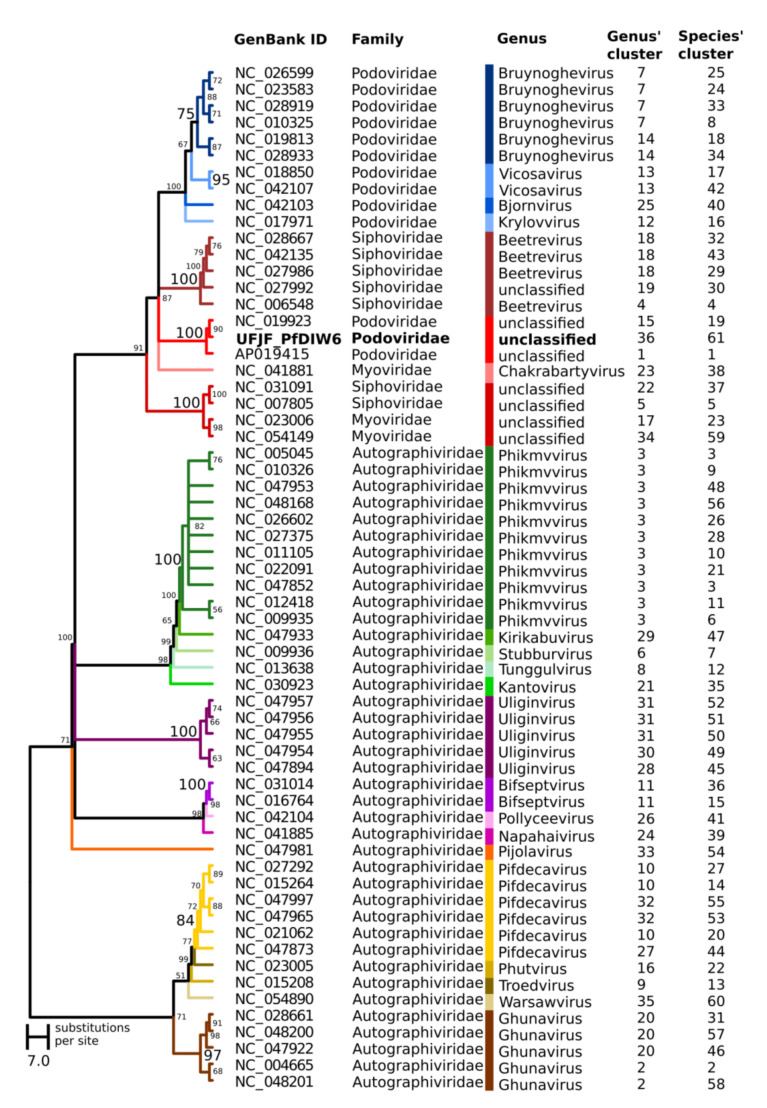
Maximum likelihood tree of the large terminase subunit of phages infecting the genus *Pseudomonas*. The midpoint rooted maximum likelihood (ML) tree was obtained by analysing terminase large subunit (TerL) amino acid sequences from UFJF_PfDIW6 and other *Pseudomonas* phages available at NCBI RefSeq database. The bootstrap values (expressed as percentages) calculated from 1000 replicates are shown beside each node. The monophyletic clades corresponding to each genus are shown in different colours. The genomes of *Pseudomonas* phages were clustered by VIRIDIC using the genomic similarity threshold of 70% for genus assignment (Genus’ cluster) and 95% for species assignment (Species’ cluster).

**Figure 3 viruses-14-00629-f003:**
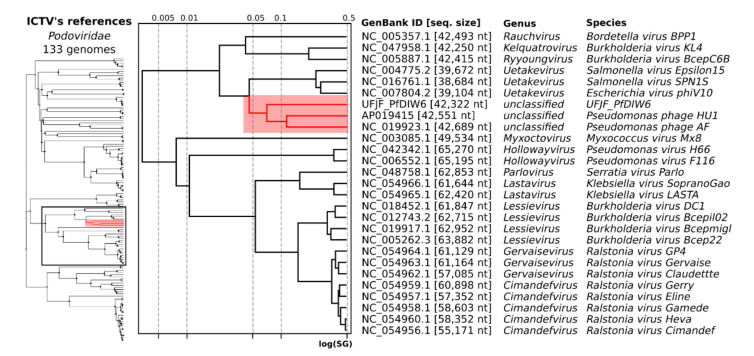
Viral proteomic tree of reference genomes for Podoviridae family. The proteomic tree was obtained from UFJF_PfDIW6 and 130 reference genomes for the genus of Podoviridae phages. The genomes were aligned all-against-all using ViPTree, and the genomic similarity scores (SG) were computed. The cluster, including UFJF_PfDIW6, is highlighted in red. The genomes of Podoviridae phages were also clustered by VIRIDIC using the genomic similarity threshold of 70% for genus assignment (Genus’ cluster) and 95% for species assignment (Species’ cluster).

## Data Availability

The data presented in this study are openly available at the NCBI (https://www.ncbi.nlm.nih.gov/genbank/). The complete genome sequence of phage UFJF_PfDIW6 is available at GenBank database, accession number OM418631.

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
