# Peer review of "Genomic Characterisation of UFJF_PfDIW6: A Novel Lytic Pseudomonas fluorescens-Phage with Potential for Biocontrol in the Dairy Industry"

_viruses, 2022, doi:10.3390/v14030629_

Round 1

Reviewer 1 Report

the study presented by Hungaro et al. presents the genomic characterization of UFJF_PfDIW6 as a novel lytic Pseudomonas fluorescent infecting phage with potential for biocontrol in the dairy industry. the bacterial -mediated spoilage of milk is an important problem in dairy industry. the use of phages has an underestimated potential to prevent this situation. 

authors aimed at the genomic description of isolated phages. however, I feel a bit unsatisfied with the introduction section. Authors should describe more precisely the use of bacteriophages in the dairy industry with the use of experimental papers, not review, mainly due that more of them concentrate on "how to avoid phage infection of lactic bacteria strains" than "how to use phages to protect milk". here are some examples to use:

https://onlinelibrary.wiley.com/doi/full/10.1111/jfs.12747

https://www.sciencedirect.com/science/article/pii/S0924224419304947#bbib39

Author Response

For the editor:

First and foremost, we would like to thank you for allowing us to revise our manuscript. We assure you that all the reviewers' comments have been taken into consideration in the revised manuscript.

Thank you very much for your comment on the proposal to abolish the order Caudovirales and, consequently, the families Myoviridae, Siphoviridae, and Podoviridae by the ICTV. We agree with this reclassification proposal of new phage families based on multiple aspects, including genome analyses. We have added a sentence in the manuscript to explain this question in more detail. Please verify lines 72-76 on page 2 and lines 418-419 on page 11.

We also agree with your comment regarding Figure 1. We have removed this figure from the manuscript and put it as supplementary material.

For reviewer #1:

We are thankful to you for giving valuable time to our manuscript. We have revised the introduction section to make it more understandable regarding your comments. We have included more details on technological problems caused by Pseudomonas in milk and dairy products. In addition, we have added studies about the use of bacteriophages to control pathogen and spoilage bacteria in the dairy industry. Please check lines 42-60 on pages 1-2.

Reviewer 2 Report

The article is reporting a novel bacteriophage, UFJF_PfDIW6, infecting Pseudomonas fluorescens. The host bacteria is a causative agent of dairy product spoilage. It is important to study and report the characterization of these phages.

The article is well written one. All the experimental data is flawless and the descriptions are also clear and clarified. 

Only a few P. fluorescens phages were reported. Thus this data would be a important one.

Author Response

We are thankful to you for giving valuable time to our manuscript. We have been performed all corrections suggested by reviewers, which may be found in the revised manuscript.

Round 2

Reviewer 1 Report

authors properly revise the manuscript

Author Response

Thank you for your considerations